# Circular SAR Incoherent 3D Imaging with a NeRF-Inspired Method

Hanqing Zhang [1,2,3], Yun Lin [4], Fei Teng [1,2,*], Shanshan Feng [1,2], Bing Yang [5] and Wen Hong [1,2]

1. Aerospace Information Research Institute, Chinese Academy of Sciences, Beijing 100094, China; zhanghanqing19@mails.ucas.ac.cn (H.Z.); fengshanshan17@mails.ucas.ac.cn (S.F.); hongwen@aircas.ac.cn (W.H.)
2. Key Laboratory of Technology in Geo-Spatial Information Processing and Application System, Chinese Academy of Sciences, Beijing 100190, China
3. The School of Electronic, Electrical and Communication Engineering, University of Chinese Academy of Sciences, Beijing 100049, China
4. School of Electronic Information Engineering, North China University of Technology, Beijing 100144, China; ylin@ncut.edu.cn
5. Beijing General Municipal Engineering Design & Research Institute Co., Ltd., Beijing 100082, China; bmedi_1@126.com
* Correspondence: tengfei@aircas.ac.cn; Tel.: +86-1358-154-3995

**Abstract:** Circular synthetic aperture radar (CSAR) has the potential to form 3D images with single-pass single-channel radar data, which is very time-efficient. This article proposes a volumetric neural renderer that utilizes CSAR 2D amplitude images to reconstruct the 3D power distribution of the imaged scene. The innovations are two-fold: Firstly, we propose a new SAR amplitude image formation model that establishes a linear mapping relationship between multi-look amplitude-squared SAR images and a real-valued 4D (spatial location $(x, y, z)$ and azimuth angle $\theta$) radar scattered field. Secondly, incorporating the proposed image formation model and SAR imaging geometry, we extend the neural radiance field (NeRF) methods to reconstruct the 4D radar scattered field using a set of 2D multi-aspect SAR images. Using real-world drone SAR data, we demonstrate our method for (1) creating realistic SAR imagery from arbitrary new viewpoints and (2) reconstructing high-precision 3D structures of the imaged scene.

**Keywords:** 3D imaging; synthetic aperture radar (SAR); circular SAR; neural radiance field (NeRF); differentiable rendering

## 1. Introduction

Circular Synthetic Aperture Radar (SAR) is a novel airborne SAR imaging mode that has gained increasing attention in recent years [1]. By flying in a circular trajectory, circular SAR (CSAR) can obtain 360° radar responses of a target, which is a significant advantage over traditional narrow-angle linear-trajectory SAR systems [2]. In the field of reconnaissance, the primary motivation for adopting a circular trajectory is to achieve continuous and comprehensive observation of complex or dynamic scenes, which can provide valuable information for target identification, classification, and tracking [3]. It is widely acknowledged that, in comparison to conventional narrow-angle linear-trajectory SAR systems, CSAR can capture more comprehensive scattering characteristics of targets [4], leading to improved accuracy in target identification and classification [5–7], enhanced detection, velocity measurement, and tracking capabilities of moving targets [8–10], as well as reduced radar blind (shadowed) areas [11], among other benefits. In addition to the aforementioned benefits that have been widely recognized, the varying range-Doppler geometry of CSAR data presents a potential for forming high-resolution 3D images in a single-flight pass [12]. This is in contrast to the tomographic SAR (TomoSAR) methods, which rely on multiple passes of data collection for creating a 3D image [13]. As a result, a

3D imaging solution with CSAR data may offer the advantage of higher time efficiency for data collection.

It is noteworthy here that the 3D imaging problems investigated in this article differ from those of multi-view stereo (MVS)-like ones. The latter typically assumes the absence of layover scatterers in SAR imagery and estimates the height of image pixels using stereo radargrammetric methods [14,15] based on this assumption. In recent years, stereo-based 3D reconstruction methods have yielded promising results with CSAR data, with some studies demonstrating dense 3D point cloud reconstruction with sub-meter accuracy [16]. However, we have to acknowledge here that layover scatterers are prevalent in SAR imagery, particularly in urban areas [17]. In contrast to stereo methods, the 3D imaging addressed in this paper focuses more on separating the layover scatterers along radar elevation directions and generating radar images with true 3D resolution.

Airborne SAR 3D imaging is typically treated as an inverse imaging problem [18]. TomoSAR is now the most commonly used 3D imaging mode for single-channel radar systems, which reconstructs the 3D reflectivity by querying the multi-baseline data in elevation directions. However, TomoSAR typically requires a significant number of repeated flight passes to improve the resolution and reduce sidelobes in elevation, which may limit its feasibility in reconnaissance applications. In airborne scenarios, the number of repeated passes is typically up to 10~20 [19]. Moreover, for drone platforms, the applicability of TomoSAR may be further compromised by motion errors and irregular baselines [20]. These factors motivate the development of the new 3D imaging mode in this article, which is expected to achieve fine 3D resolution from single-channel single-pass SAR data and can be easily deployed with drone platforms.

The CSAR imaging mode offers the potential to achieve fine 3D resolution without the need for interferometric baselines. To the best of our knowledge, almost all current 3D imaging methods for CSAR utilize the coherent imaging strategy to create a 3D image [21]. Specifically, the standard coherent 3D imaging method for CSAR is the filtered back-projection method, where all radar echoes are coherently added to a 3D image grid [22]. (However, as we will explain in the next paragraph, this method may not be a wise choice for drone-collected data.) Coherent processing with ideal isotropic point scatterers theoretically provide a very high horizontal resolution up to the order of radar wavelength and a vertical resolution on the order of radar bandwidth [23]. The achieved 3D resolution of coherent imaging methods has been verified with real airborne data on man-made targets at X-band [24] and natural targets at L-band [22]. When CSAR is utilized as a wide-angle coherent imaging system, the anticipated 3D resolution as a function of the processed azimuth aperture size has been discussed in previous studies [25], revealing that an azimuth aperture with an angular extent of approximately 45° would yield an elevation resolution that is comparable to the radar bandwidth.

However, for lightweight drone platforms, coherent imaging over such a large curved aperture presents significant challenges. Firstly, forming a single synthetic aperture (~45° azimuth extent) typically takes several minutes, during which time the accuracy of radar position recordings should not have an error greater than 1/4 of the radar wavelength; otherwise, the quality of the 3D image may be degraded [26]. To address this issue, in almost all published airborne CSAR experiments, special artificial reflectors need to be placed in the scene to realize the correction of residual motion errors [22,24]. However, this approach can only be achieved in cooperative scenarios. Many drone SAR systems are now only equipped with tactical-grade navigation systems, which do not easily meet the requirements for 3D coherent processing. Secondly, the wide-angle coherent imaging method requires the scatterers to conform to the isotropic point scattering model within a single processed aperture. This requires the inherent phase of the target's complex reflectivity to remain constant over the observation angle changes of several tens of degrees [2]. However, many targets in natural and urban scenes are so-called distributed scatters. Even a slight change in the radar viewing angle can make a significant phase shift in their complex reflectivity. Such

targets are inherently unsuitable for obtaining high 3D resolution through the wide-angle coherent imaging methods.

Aforementioned issues with a coherent 3D inversion method have motivated us to develop the incoherent 3D imaging strategy in this article. In our method, we first utilize the sub-aperture imaging strategy to process the CSAR data into a series of 2D sub-aperture images. Then, we only use the 2D amplitude images to deduce the target's 3D scattering power distribution. This 3D inversion problem is formulated as an inverse imaging problem. To address this inverse problem, we first build a forward image formation model in Section 2, which elucidates the relationship between the 2D amplitude images and the target's 4D (spatial coordinates $(x, y, z)$ and radar azimuth angle $\theta$) reflectivity distribution. In this section, we adopt a random volume scattering assumption for radar scatterers to make the image formation model sufficiently differentiable, which enables us to solve the inverse imaging problems using differentiable rendering techniques. Specifically, our fundamental assumption is that the target's complex reflectivity should roughly conform to Goodman's volume scattering model [27]. Based on this assumption, we establish a linear mapping relationship between the target's 4D scattering energy distribution and the multi-look amplitude-squared SAR images, which is differentiable.

In Section 3, we introduce our second innovation, which involves extending the Neural Radiance Field (NeRF) [28] to solve the 4D inverse imaging problem raised in Section 2. NeRFs have gained significant popularity in the field of differentiable volume rendering. NeRF uses multi-layer perceptrons (MLPs) to fit the target's radiance fields, which can create virtual but realistic new perspectives as well as learning the target's 3D shape. So far, some methods based on NeRF transformations have achieved remarkable results in the problems that have close similarities to the CSAR inverse imaging problems raised in Section 2. These include extending NeRF to solve sparse-view computed tomography (CT) reconstruction problems [29], 3D imaging problems with imaging sonar data [30], and non-line-of-sight (NLOS) imaging problems with 1D light transient collections [31], among others. As the image formation model of SAR proposed in Section 2 is similar to that of imaging sonar, CT, and time-of-flight (ToF)-based NLOS imaging problems, we shall solve the non-convex optimization problem of CSAR 4D inverse imaging by adapting the NeRF model. Specifically, considering the SAR phenomenology and geometry, we redefine the concept of 'rays' in the NeRF models, redesign the rendering equations, and redesign new error functions. Above aspects will be discussed in detail in Section 3.

In Section 4, we present a demonstration of the performance of our proposed method using real-world CSAR data collected by a drone platform. We showcase our method's ability to generate new viewpoints and perform 3D reconstruction. We also quantitatively analyze the height accuracy of the learned 3D model. In Section 5, we provide a brief conclusion of this article. The primary goal of this article is to present a simple yet efficient SAR 3D imaging solution. The advantages of our method may be (1) using single-channel single-pass data for 3D imaging, which improves the data acquisition efficiency, and (2) utilizing an incoherent 3D imaging strategy for the circular SAR data, which does not require high-precision motion recordings, thus enabling high-quality 3D imaging with drone SAR systems. (3) Additionally, inspired by NeRF, we first introduce the MLP to solve the non-convex optimization problem of CSAR scene-level 3D inversion.

## 2. SAR Amplitude Image Formation Model

The 3-D reconstruction problem in this article is an inverse imaging problem. In this section, we aim to build a forward image formation model for CSAR 3D (or more accurately, 4D) inversion problems. The forward image formation model should be differentiable mathematical expressions that account for the generation of low-dimensional observations, typically 2D SAR images, from the unknown high-dimensional functions. In this article, our incoherent 3D imaging method is to reconstruct the target's 4D reflectivity distribution using multi-aspect SAR 2D amplitude images. (Although raw SAR images are complex-valued, the phase component of image pixels is typically considered as random

noises. Therefore, we only utilize SAR amplitude images to realize 3D imaging, which fundamentally differs from the TomoSAR 3D inversion problems.)

In this section, we derive a mathematical formula that accounts for the generation of SAR 2D amplitude images. Our fundamental assumption is that the complex reflectivity of all radar scatterers should approximately conform a certain volume scattering model [27]. Under this assumption, we establish a linear mapping relationship between the multi-aspect SAR amplitude images and a real-valued 4D (spatial location $(x, y, z)$ and radar azimuth angle $\theta$) scattered field of the target. The image formation model deduced in this section is inspired by those used in incoherent tomography techniques [32] and multi-master SAR tomography techniques [33,34], while the neural network in the next section is used to solve the inverse problem of this image formation model.

The CSAR imaging geometry is illustrated in Figure 1a. We employ the sub-aperture imaging method to segment and process the CSAR echo data into a set of 2D images. Specifically, we divide the 360° circular aperture into a series of small sub-apertures. When computing the range-Doppler geo-location model for radargrammetry applications, each sub-aperture will be approximated as a straight line one, as suggested in [15]. Then, conventional SAR imaging methods will be applied to each sub-aperture to create a sequence of complex-valued SAR images $I_1, I_2, \ldots, I_N$.

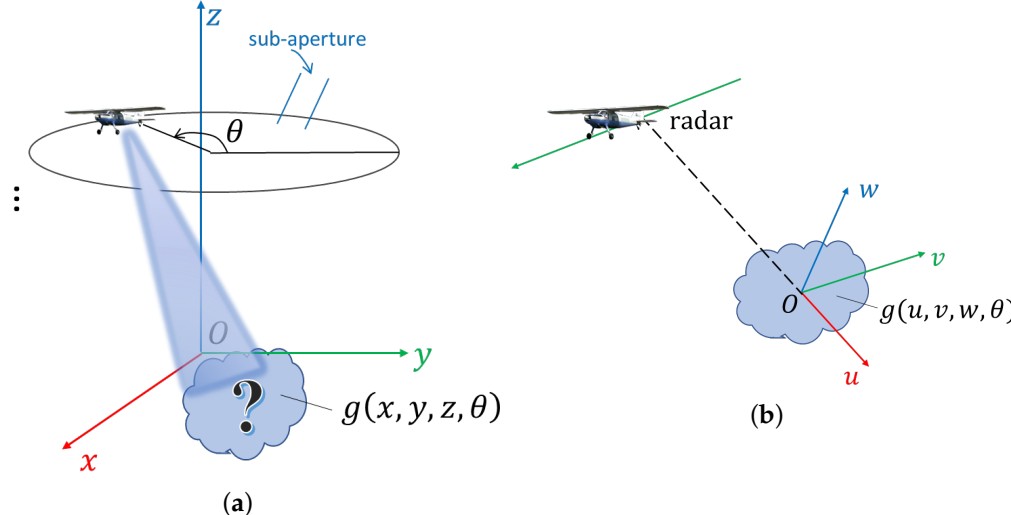

(**a**)

(**b**)

**Figure 1.** CSAR imaging geometry. (**a**) Full aperture imaging geometry. (**b**) Sub-aperture imaging geometry.

As a single sub-aperture can be approximated as a straight line, the SAR imaging geometry of a single sub-aperture can be approximated as in Figure 1b. Here, $g(x, y, z)$ (or $g(u, v, w)$) represents the 3D complex-valued reflectivity of the target. (We assume for now that the complex scattering properties of the target are independent of the radar aspect angles $\theta$). In the figure, $(u, v, w)$ is a 3D Cartesian coordinate system, where the three axes represent the range, azimuth, and elevation dimensions of the SAR image, respectively.

We can use the direction of radar wavenumber vector $\boldsymbol{k}$ to distinguish the SAR images collected at different aspect angles $\theta$. In this manner, the 2D complex images can be uniformly represented as [35]

$$\boldsymbol{I} = I(u, v; \boldsymbol{k}) = \boldsymbol{s_A} \otimes \left\{ \int_w g(\boldsymbol{r}) e^{-j\langle \boldsymbol{r} - \boldsymbol{r}_{APC}, \boldsymbol{k} \rangle} \mathrm{d}w \right\} \tag{1}$$

where $\boldsymbol{r} = (u, v, w)$ denotes the 3D spatial coordinates, $\otimes$ represents the convolution operation, vector $\boldsymbol{r}_{APC}$ denotes the location of azimuth aperture center, vector $\boldsymbol{k}$ denotes the round-trip radar wavenumber, $\langle \cdot, \cdot \rangle$ denotes the inner product operation of two vectors,

and $s_A(\cdot)$ is a 2D *sinc* function representing the point spread function (PSF) in 2D SAR images, with

$$s_A(u,v) = B_u B_v \operatorname{sinc}\left(\frac{uB_u}{2\pi}\right) \operatorname{sinc}\left(\frac{vB_v}{2\pi}\right) \tag{2}$$

Here, $B_u$ and $B_v$ are the bandwidth of SAR images along $\boldsymbol{u}$ and $\boldsymbol{v}$ directions, respectively. $B_u$ and $B_v$ are typically close to each other. Function $s_A(\cdot)$ is also referred to as the aperture function, as it represents the bandwidth of the 2D synthetic aperture.

To remove the phase component from an image, we square the amplitude values of each pixel and discard its phase component. This is actually equivalent to an autocorrelation operation. For the sake of clarity, we temporarily ignore the impact of the aperture function $s_A(\cdot)$. The resulting image from this operation can be represented as

$$I\bar{I} = I(u,v;\boldsymbol{k})\overline{I(u,v;\boldsymbol{k})} \approx \iint g(u,v,w)\overline{g(u,v,w')}e^{-jk_w(w-w')}\mathrm{d}w\,\mathrm{d}w' \tag{3}$$

Here, the horizontal line above the variables indicates the complex conjugate operation. The variable $k_w$ represents the wavenumber vector component along the $\boldsymbol{w}$ direction.

The double integral in the above equation may seem too complicated for a differentiable rendering solver, so we adopt the following statistical assumption to simply it. We assume that the 3D complex reflectivity function $g(\boldsymbol{r})$ approximately follows Goodman's scattering model [27], which implies that each scattering unit of $g(\cdot)$ follows a zeros-mean complex white Gaussian distribution:

$$g(\boldsymbol{r}) \sim \mathcal{CN}\left(0, \frac{\sigma^2(\boldsymbol{r})}{2}\right) \tag{4}$$

Given that $g(\boldsymbol{r})$ is white, it follows that its power spectral density is stationary. We now examine the expectation of the interferogram generated by the multi-looking process. (For SAR images, multi-looking roughly performs the function of spatial averaging.) The expectation of the interferogram, denoted as $\mathcal{I} = \mathrm{E}[I\bar{I}]$, can be viewed as a sample from the autocorrelation function of $I(u,v;\boldsymbol{k})$ evaluated at the spatial frequency location $\Delta\boldsymbol{k} = \boldsymbol{0}$.

$$\mathcal{I}(u,v) = \mathrm{E}[I(u,v;\boldsymbol{k})\overline{I(u,v;\boldsymbol{k})}] = R_{II}(u,v;\Delta\boldsymbol{k})|_{\Delta\boldsymbol{k}=\boldsymbol{0}} \tag{5}$$

Here, $R_{II}(u,v;\Delta\boldsymbol{k})$ represents the autocorrelation function of Image $I$.

According to the Van-Cittert and Zernike theorem, $R_{II}(u,v;\boldsymbol{0})$ is equal to the Fourier Transform of the 3D backscatter coefficient function $\sigma(u,v,w) \in \mathbb{R}_+$ evaluated at $\Delta\boldsymbol{k} = \boldsymbol{0}$, which means

$$\mathcal{I}(u,v) = R_{II}(u,v;\boldsymbol{0}) = \int \sigma(u,v,w)\mathrm{d}w \tag{6}$$

In the following content, we will use the term 'radar scattered field' to refer to the function $\sigma(\cdot)$. It should be noted that $\sigma(\cdot)$ is assumed to be real-valued according to the assumption made in Equation (2).

In above derivations, we did not consider the effect of the aperture function $s_A(\cdot)$, while the actual images are band-limited. For simplicity, we approximate the complete expression of $\mathcal{I}$ as

$$\mathcal{I}(u,v) \approx \boldsymbol{s'_A} \otimes \int \sigma(u,v,w)\mathrm{d}w \tag{7}$$

Obtaining an exact formula for the aperture function $s'_A(\cdot)$ may pose a challenge. To simplify matters, we shall assume that $s'_A(\cdot)$ has the same expression with the original aperture function $s_A(\cdot)$, i.e., $s'_A(\cdot) \simeq s_A(\cdot)$.

Furthermore, we take into account the anisotropic nature of the observed targets and add an additional dimension to the function $\sigma(\cdot)$, making it a 4D field $\sigma(u, v, w, \theta)$. The final forward image model is

$$\mathcal{I}_\theta(u, v) \approx s_A \otimes \int \sigma(u, v, w, \theta) \mathrm{d}w \qquad (8)$$

Here, $\sigma(u, v, w, \theta)$ is a real-valued function that is referred to as the radar scattered field and is also the parameter of the Goodman scattering model in Equation (2). $\mathcal{I}_\theta$ is the SAR image obtained after squaring and multi-looking processing, collected from the azimuth angle of $\theta$.

Equation (8) is the final forward image formation model, and its inverse problem is a 4D inversion problem.

$$\sigma(u, v, w, \theta) \xrightarrow{\text{Equation (8)}} \{\mathcal{I}_\theta(u, v)\} \xrightarrow{\text{inversion}} \hat{\sigma}(u, v, w, \theta)$$

Therefore, a simple flow chart for the 3D reconstruction method proposed in this article can be described as shown in Figure 2.

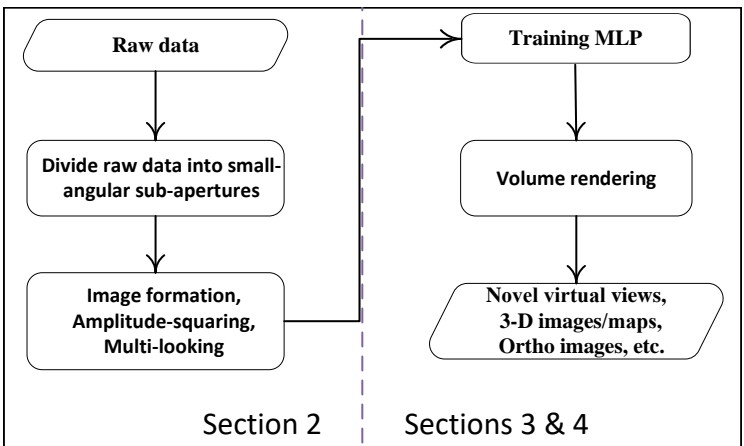

**Figure 2.** A flowchart for 3D imaging with CSAR data.

## 3. A NeRF-Inspired Network for CSAR 3D Inversion

In this section, we extend the NeRF method to solve the CSAR 4D inverse imaging problem presented in the previous section. Our proposed method shares similarities with the original NeRF method [28] in that both adopt an implicit 3D representation and employ MLP as the optimizer. However, there are several major differences. The volume rendering method used in the original NeRF method is not directly applicable to SAR images. Therefore, we derive a novel volume rendering method based on our derivation in Section 2. And, the 3D scattering energy is accumulated to the SAR image plane along the elevation direction rather than the line-of-sight (LOS) direction. Because SAR phenomenology and geometry differ from optical cameras, we build a new volumetric rendering pipeline based on elevation 'rays'.

In this section, we model the scene to be solved as a 4D scattered field $\sigma(x, y, z, \theta)$, which is (1) real-valued, (2) continuous in all dimensions, and (3) penetrable by radar waves with no energy attenuation. Here, the first condition is derived from Goodman's volume scattering model (cf. Section 2). The second condition is a fundamental assumption of NeRF. And the last condition states that we will not explicitly model occlusions in the scene. If a scatterer is not visible from a certain aspect angle, our model shall assume that its scattering energy goes to zero, rather than being occluded by a surface that radar waves cannot penetrate.

### 3.1. Network Architecture

Our neural network, as shown in Figure 3, is developed based on Instant-NGP [36], which is a NeRF variation with a significantly faster solving speed than the original NeRF model. The architecture of our network is very similar to that of Instant-NGP. However, the meanings and usages of the inputs and outputs of this network differ from those in the original NeRF methods.

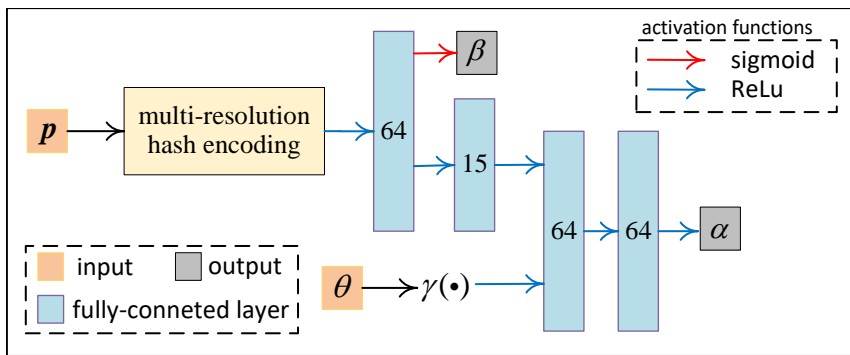

**Figure 3.** Network architecture overview.

The inputs are:

-$p$: A 3-valued vector representing the spatial coordinates of a 3D point (on a traced ray).

-$\theta$: $\theta \in [-\pi, \pi]$ is the radar aspect angle, and $\gamma(\cdot)$ in Figure 3 is the positional encoding function proposed in the original NeRF paper [28], which generates a vector of length $2L$ in the following manner:

$$\gamma(\theta) = \left(\sin\left(2^0\theta\right), \cos\left(2^0\theta\right), \ldots, \sin\left(2^{L-1}\theta\right), \cos\left(2^{L-1}\theta\right)\right) \tag{9}$$

We set L = 6 in this article.

The neural network part formulates a volumetric function, which writes $\mathcal{F} : (x, \theta) \rightarrow (\beta, a)$. The outputs are

-$\beta$: A scalar in range $[0, 1]$ that is independent of radar aspect angle $\theta$. $\beta$ represents the probability of the existence of a scatterer at spatial location $p$. In our methodology, we encourage multi-view consistency by predicting $\beta$ solely based on the spatial coordinate $p$, as well as integrating other priors with this variable.

-$\alpha$: A non-negative scalar representing the scattered energy density at 3D location $p$ and radar azimuth angle $\theta$. $\alpha$ can be considered as the predicted energy density of the unknown scattered field $\sigma(p, \theta)$.

In Figure 3, we use the multi-resolution hash encoding technique employed in the original Instant-NGP paper to encode the vector $p$. The network structure of this part is taken directly from Instant-NGP.

### 3.2. Volume Rendering with the Network

Given a set of multi-view images, the training of a NeRF model is based on rendering the intensities along individual rays traced across the scene and projected onto the pixels. In this subsection, we introduce our volume rendering strategy for SAR images. To this end, we need to trace a ray that traverses through the 3D volume of interest, query the NeRF model, and render the ray using a designated rendering function. This process is illustrated in Figure 4.

Due to the distinct SAR phenomenology and geometry, our ray tracing and rendering strategy slightly differs from that of the original NeRF method. Nevertheless, the guiding principle behind this process still adheres to the fundamental concept of differentiable rendering.

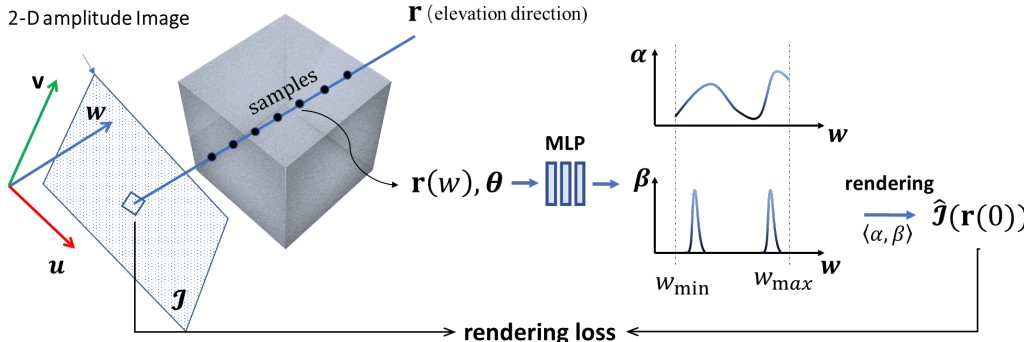

**Figure 4.** Overview of the training process. The key difference compared to the original NeRF model is that our forward model is built on 'elevation' rays.

### 3.2.1. Definition of a Ray

In the original NeRF model, rays are created along the line-of-sight (LOS) direction. However, in the SAR imaging model, scatterers in 3D space are projected onto the 2D image along the approximate elevation direction $w$ (more specifically, the same direction with the iso-range-iso-Doppler lines), rather than the LOS direction (cf. Equation (8). Therefore, we adopt the Born approximation and redefine the ray $\mathbf{r}$ as a straight-line segment passing through the scene along the direction of $w$ (cf. Figure 4). The 3D coordinates of a point $\boldsymbol{p}$ on the ray $\boldsymbol{r}$ are determined by its $w$-axis coordinate, i.e., $\boldsymbol{p} = \mathbf{r}(w)$.

For tracing a ray, two different Cartesian coordinate systems are utilized. The $(u, v, w)$ frame is defined by the radar's range-Doppler geometry that varies with the radar aspect angle $\theta$. The $(x, y, z)$ frame is a fixed coordinate system, while the coordinate of $\boldsymbol{p}$ is taken from this system. The transformation between the two frames is an affine transformation. The $(x, y, z)$ coordinate system can be chosen arbitrarily (we adopt the South-East-Up (SEU) coordinate frame in this paper.).

PS: Unlike the original NeRF method, our method does not require setting a direction for $\mathbf{r}$. This is because, in our method, we do not consider the energy attenuation along a ray, and all spatial locations are assumed to receive the same incident energy.

### 3.2.2. Rendering a Ray

For training a NeRF network, we need to redefine the manner in which the volume rendering of a ray is performed. The rendering process will convert the network output to the prediction of the pixel values in an SAR image:

$$(\mathbf{r}(w), \theta) \xrightarrow{\text{MLP}} (\alpha, \beta) \xrightarrow{\text{renderer}} \hat{\mathcal{I}}_\theta(\mathbf{r_0})$$

Here, $\mathbf{r_0} = \mathbf{r}(0)$ represents the coordinates of the intersection point between $\mathbf{r}$ and the SAR image plane, and $w_{min}$ and $w_{max}$ represent the $w$-coordinates of the two intersection points between the ray $\mathbf{r}$ and the bounding box of the interested scene.

For NeRF, the volume rendering function needs to have good differentiability properties. Based on the forward model proposed in section 2, we render the ray in the following manner. The predicted value for the ray is generated by an integrand:

$$\hat{\mathcal{I}}_\theta(\mathbf{r_0}) = \rho_{\text{rg}}\rho_{\text{az}} \int_{w_{\min}}^{w_{\max}} \beta(\mathbf{r}(w))\alpha(\mathbf{r}(w), \theta)\mathrm{d}w \tag{10}$$

Here, $\rho_{rg}$ and $\rho_{az}$ are the range and azimuth resolutions of the SAR imagery. The above equation integrates the energy density function $\alpha$ weighted by $\beta$ along the $w$ direction to obtain an estimate of the intensity value of an image pixel. The above equation is modified from the forward imaging model derived in Equation (8).

We numerically estimate the continuous integral in Equation (10) using quadrature. We divide $[w_{min}, w_{max}]$ into $M$ evenly-spaced bins and take one sample in each bin to estimate the $\hat{\mathcal{I}}(\mathbf{r_0})$ as

$$\hat{\mathcal{I}}(\mathbf{r_0}) = \rho_{\text{rg}}\rho_{\text{az}} \sum_{i=1}^{M} \beta_i \alpha_i \delta_i \tag{11}$$

where $\delta_i = |w_{i+1} - w_i|$ is the distance between adjacent samples.

Below, we construct the loss functions and train this model following the methods of NeRF and its variants.

### 3.3. Training Loss

Our loss function is comprised of two parts. The first part is the sum of squared error between the rendered and true pixel intensities of image $\mathcal{I}$. The rendering loss is

$$\mathcal{L}_{\text{int}} \equiv \sum_{\mathbf{r}} \left\| \log(\hat{\mathcal{I}}(\mathbf{r_0})) - \log(\mathcal{I}(\mathbf{r_0})) \right\|_2^2 \tag{12}$$

In the above equation, we compute the error using the logarithmic values of the pixel. This is because the speckle noise in SAR images behaves more like multiplicative noise rather than additive noise.

The second part of the loss function is based on our prior knowledge and is used to encourage sparsity. We first add an $\ell_1$ loss term:

$$\mathcal{L}_{\text{reg1}} \equiv \frac{1}{M} \sum_{\mathbf{r}(w)} \alpha(\mathbf{r}(w)) \tag{13}$$

Then, we add another loss term that aims to encourage the values of $\alpha$ to be either 0 or 1, which is

$$\mathcal{L}_{\text{reg2}} \equiv -\frac{1}{M} \sum_{\mathbf{r}(w)} \alpha(\mathbf{r}(w)) \log(\alpha(\mathbf{r}(w))) \tag{14}$$

Hence, our final training loss term is

$$\mathcal{L} = \mathcal{L}_{\text{int}} + \lambda_{\text{reg }1}\mathcal{L}_{\text{reg }1} + \lambda_{\text{reg }2}\mathcal{L}_{\text{reg }2} \tag{15}$$

## 4. Experiments

### 4.1. Circular SAR Campaign

The CSAR data used in this section were collected in 2020 with an experimental W-band single-channel radar system developed by the Nanjing Research Institute of Electronics Technology (NRIET).

A photo of the radar platform and a schematic of the flight trajectory for data collection are given in Figure 5. The radar platform is a small fixed-wing drone that is integrated with a GPS chip, which enables a horizontal positioning accuracy of about 0.5 m.

The flight trajectory was originally designed as a complete circular path, with the platform flying at a height of approximately 500 m above the ground level and a flight radius of approximately 1000 m. However, due to the limitation of the on-board memory capacity, only echoes within 3/4 of the circular path were recorded. The observed scene is located about 200 m apart from the center of the circular path, and the diameter of the area that is continuously illuminated by the radar beam is approximately 100 m. (It should be emphasized that the flight trajectory does not need to strictly follow a standard circular path, as the algorithm proposed in this paper can be applied with SAR data obtained from most curved trajectories).

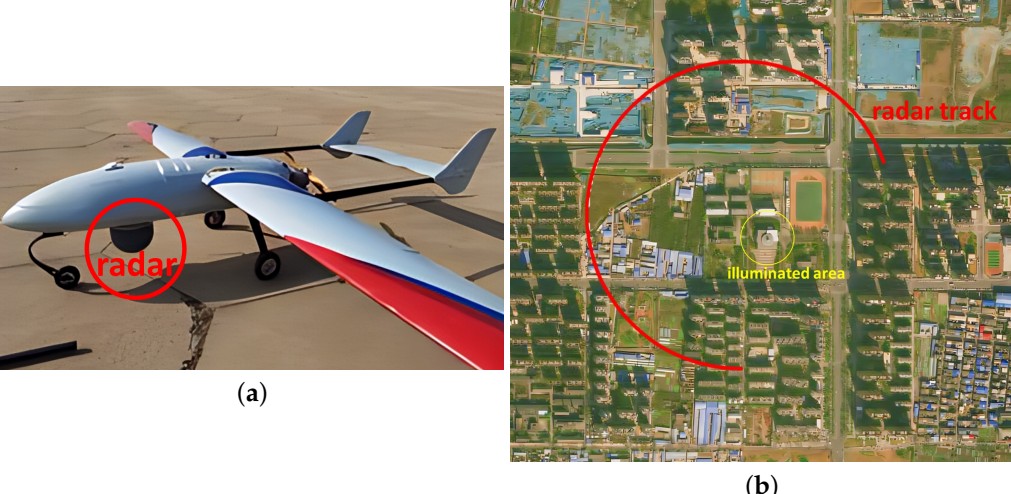

**Figure 5.** Experiement setup and CSAR imaging geometry. (**a**) A photo of the drone, the platform, and the radar device. (**b**) CSAR imaging geometry (red colored curve).

A photo of the observed scene is given in Figure 6a. The illuminated area is a school located in Henan province, China. The radar spot center is set as a teaching building that is approximately four stories high and has a square footprint. After data collection, we cut the CSAR data into a series of non-overlapping sub-apertures, with the image azimuth resolution being approximately 0.25 m. Figure 6b shows one sub-aperture image, whose resolution is approximately 0.25 m × 0.25 m. Other parameters of the radar system and data processing parameters are given in Table 1.

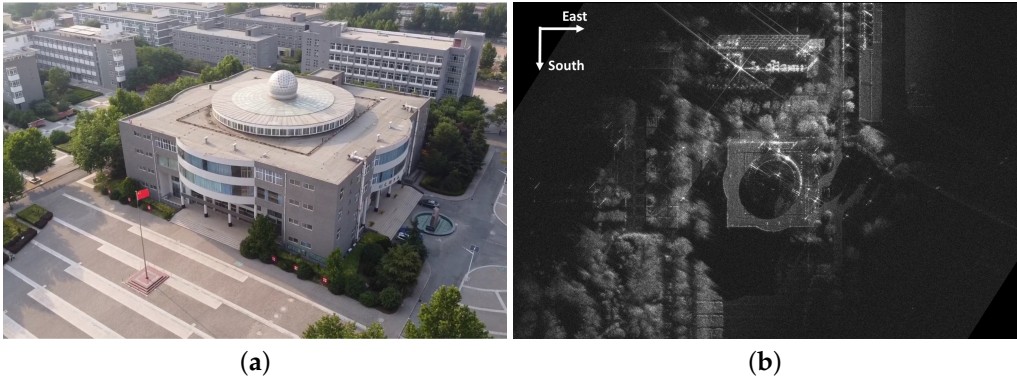

**Figure 6.** Sample SAR image and optical photo of the imaging area. (**a**) Optical photo, (**b**) SAR imagery.

**Table 1.** Acquisition and processing parameters.

| Parameter | Description |
| --- | --- |
| Center frequency | 94 GHz |
| Radar power | <0.5 watt |
| Bandwidth | up to 900 MHz |
| Flight height | 500 m AGL |
| Flight radius | 1000 m |
| Incidence angle | 65° |
| sub-aperture image resolution (slant range × azimuth) | 0.25 m × 0.25 m |

### 4.2. 3D Imaging Experiments

We then used the proposed method to estimate the 4D scattered field $\sigma(x, y, z, \theta)$.

Firstly, we performed the amplitude-squaring and multi-looking process for the SAR amplitude images, following the pipeline outlined in Figure 2, to generate the training images $\mathcal{I}_\theta(x, y)$. We uniformly select 55 images from the total 270° azimuth aperture, with a 5-degree azimuth angle difference between adjacent images. These 55 images were used as training samples for the Network.

The purpose of the neural network is to learn the 4D scattered field $\sigma(x, y, z, \theta)$ of the imaged scene. The axis-aligned bounding box of the observed scene is set to be centered at the teaching building, with a size of 350 m $\times$ 350 m $\times$ 30 m (south-east-up). For the multi-resolution hash encoding layers, we used the default parameters of Instant-NGP for an RTX 2080 GPU. The finest resolution was set to 512, and Hash table size was set to $2^{19}$. For training our network, we considered a batch size of 1024 rays and employed $M = 256$ uniform samples along each ray. We depot the Adam optimizer with hyper parameters $\beta_1 = 0.9$ and $\epsilon = 1 \times 10^{-7}$, and we used a learning rate that started at $5 \times 10^{-4}$. The training of this model takes 4 epochs to converge, resulting in $\sim$20 min on an NVIDIA RTX 2080 GPU with 11 GB RAM.

The 3D images obtained by the network are shown on the left side of Figure 7, where the color map encodes the height value (along the z-direction) of the scattered energy. The images on the left side of Figure 7 were generated using the volume rendering techniques with a pixel resolution of 0.5 m $\times$ 0.5 m. For comparison, real SAR images obtained from similar viewpoints are given on the right side of Figure 7 (these viewpoints were not used as training data for the network).

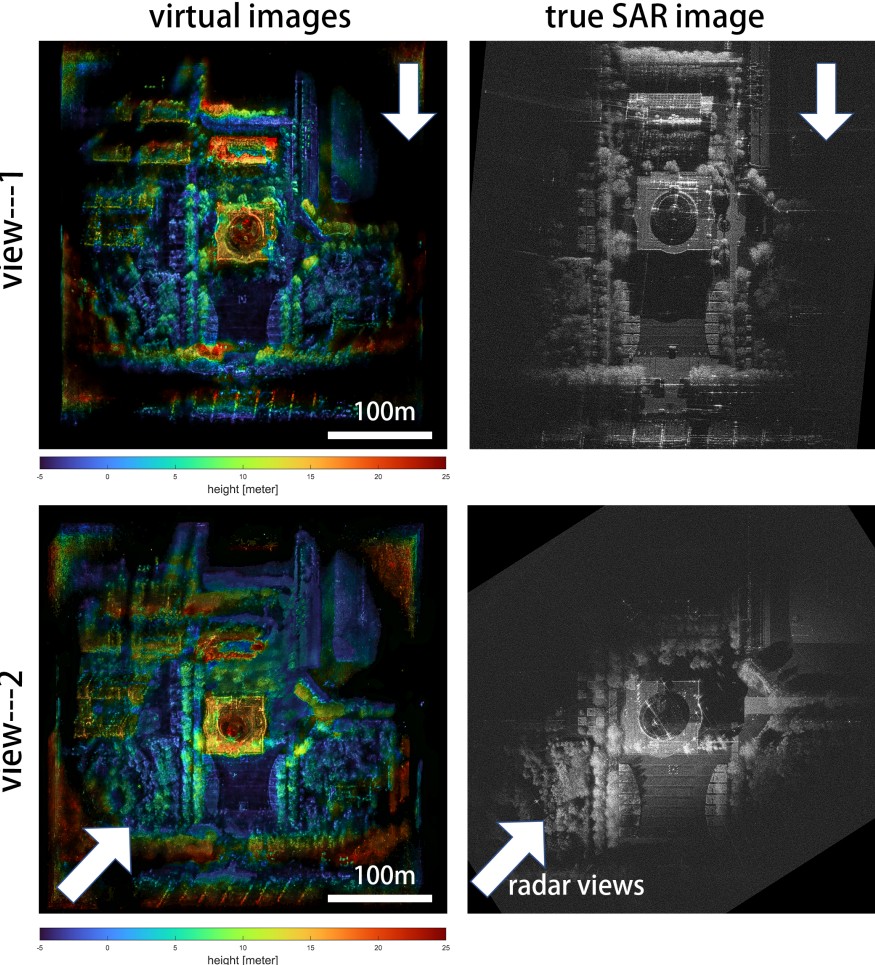

**Figure 7.** 3D processing result. The left images are rendered by the neural network. The images on the right are real SAR 2D images, presented for comparison with the ones on the left. The arrows in the images indicate the radar viewing direction.

The network can realize two different functions: one is to create virtual SAR images from arbitrary viewpoints, and the other is to learn the 3D structures of the scene. For generating SAR images from new viewpoints, the image on the left side of Figure 7 is output by the neural network, while the image on the right side is the ground-truth data. Comparing the two images, it is believed that the neural network can output highly realistic SAR images.

The neural network can also generate new viewpoints that were not captured. For example, in our data collection task, we only recorded data from 3/4 of the circular aperture, and data from the other 90 degrees of azimuth aperture were not recorded due to storage capacity limitations. Using the trained MLP, we now can make an estimation for theses missing data. In Figure 8, we rendered an SAR image from a new viewpoint, the observation geometry of which is given in Figure 8a. The nearest neighborhood viewpoint to this virtual image has an azimuth angle difference of up to 45 degrees from it. However, as shown in Figure 8b, the generated virtual image shows no significant geometric distortions (at least in the central area), indicating that the trained NeRF may have successfully learned the 3D structures of the scene.

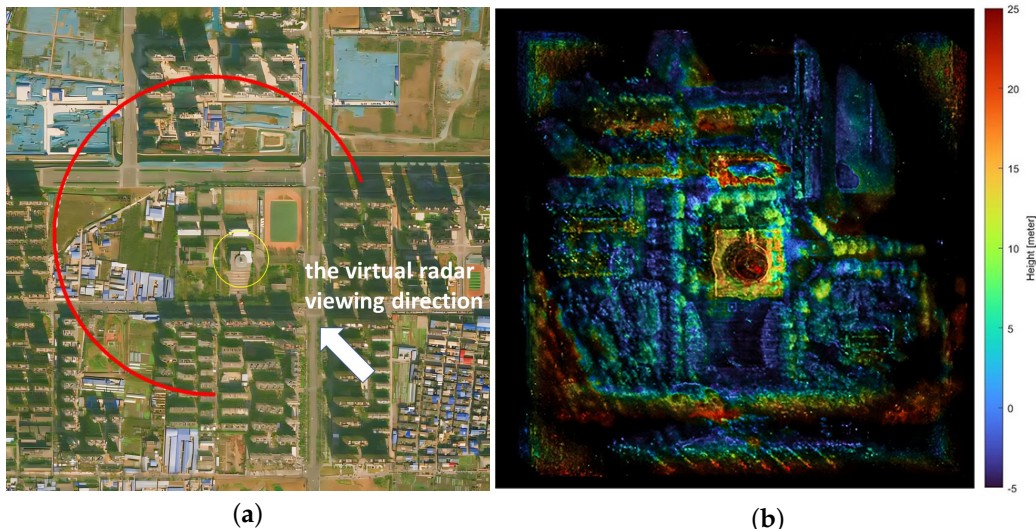

(a)      (b)

**Figure 8.** NeRF-rendered image under a virtual SAR imaging geometry. (**a**) Schematic diagram of the virtual SAR viewing direction. (**b**) SAR image created using volume rendering with the trained NeRF.

Furthermore, as shown in Figure 9a, we used the trained model to generate an orthorectified SAR image, which is equivalent to projecting the scene onto the horizontal plane. In this image, the vertical structures of the scene, such as the facades of the buildings, are projected into the buildings' footprints. Even real SAR cannot collect such a viewpoint directly—it can be generated by a trained NeRF model. In Figure 9b, we provide a vertical slice taken along the white dashed line in Figure 9a. Through the slicing operation, the scattering energy distribution of the target along the vertical direction estimated by the neural network can be visualized more precisely. We can observe the vertical distribution of scattered energy from buildings, trees, cement floors, and other objects in the image.

The primary objective of NeRF is to learn the 3D structures of the scene. Therefore, our primary concern is to evaluate the geometric accuracy of the learned 3D model. This requires the ground truth data of this area's 3D structure. However, we did not have the equipment to collect such ground truth data, and we did not find any available high-precision 3D maps of this area either. Nevertheless, in our previous work [37], we proposed an 'MVS'-like method for reconstructing high-precision 3D point clouds with multi-aspect SAR images and demonstrated that this method can create 3D points with a height accuracy better than 1 m with SAR images at a resolution of 0.3 m. According to studies on the original NeRF method, the 3D positioning accuracy of the NeRF method is typically inferior to MVS methods like Colmap [38] (the advantage of NeRF lies in the high visual quality

and completeness of the reconstructed 3D model). Therefore, in this paper, we first used the stereo SAR method to reconstruct a 3D point cloud of the scene, and then we used the reconstructed points to evaluate the accuracy of the 3D structure learned by NeRF. The stereo point cloud reconstruction was performed using the method proposed in [37], with a maximum azimuth angle difference of 30° between stereo image pairs. Using this algorithm, we reconstructed approximately 5.4 million 3D points for the scene, as illustrated in Figure 10.

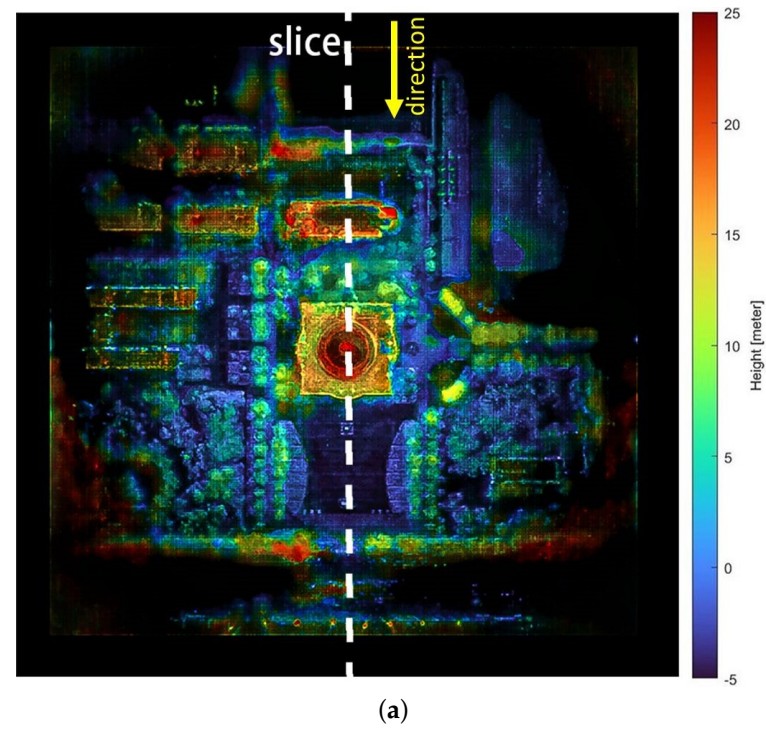

(**a**)

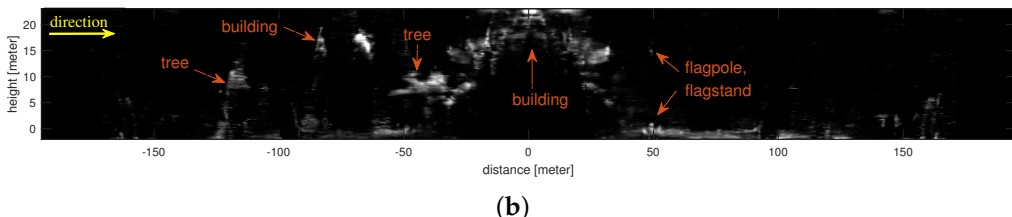

(**b**)

**Figure 9.** Visual assessment of 3D imaging results. (**a**) Ortho image of the scene, created by NeRF. (**b**) A vertical slice along the white dashed line in (**a**).

We adopted these stereo points as the test data. Here, we assumed that most areas of the scene do not have layover pixels. Based on this assumption, we employed the following volume rendering method to query the trained NeRF model and generate estimated heights for these stereo points:

$$\hat{h} = \frac{\sum_{i=1}^{M} \beta_i \alpha_i \delta_i h_i}{\sum_{i=1}^{M} \beta_i \alpha_i \delta_i} \tag{16}$$

Here, $h_i$ represents the height of the $i$-th sampling point on the ray.

We computed the difference $\delta h$ between the predicted height $\hat{h}$ and the true height (stereo measurements) of the 3D points. The resulting error curve is given in Figure 11.

According to Figure 11b, the overall accuracy of the height reconstruction of the scene can be estimated to be at the meter level. Meanwhile, we calculated the mean deviation between the estimated value $\hat{h}$ and the true value, which was calculated as $\mathbb{E}[\delta h] = -0.8916$ m.

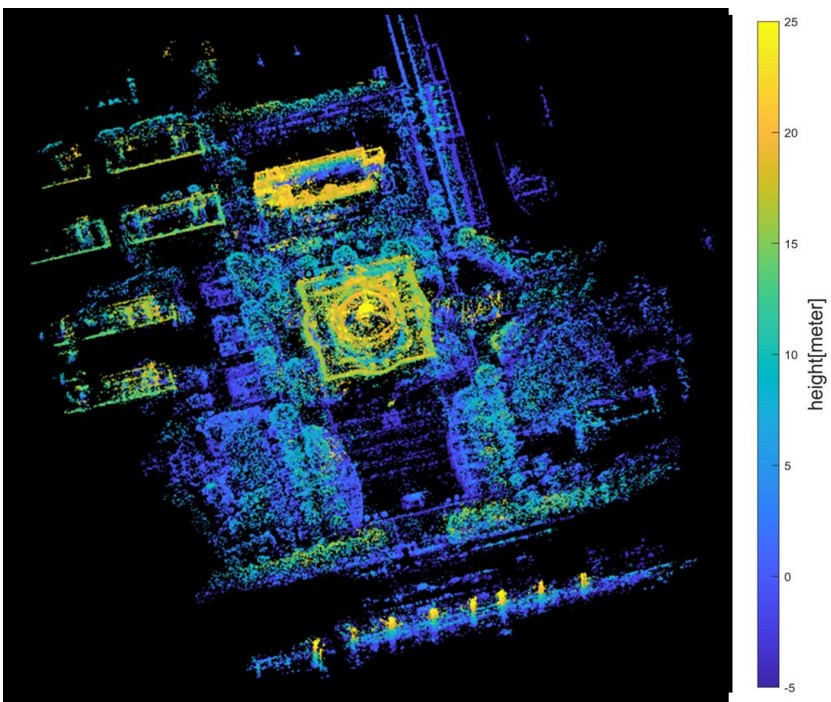

**Figure 10.** 3D point could be created by stereo SAR method. These points will be used to evaluate the accuracy of NeRF reconstructed 3D structure.

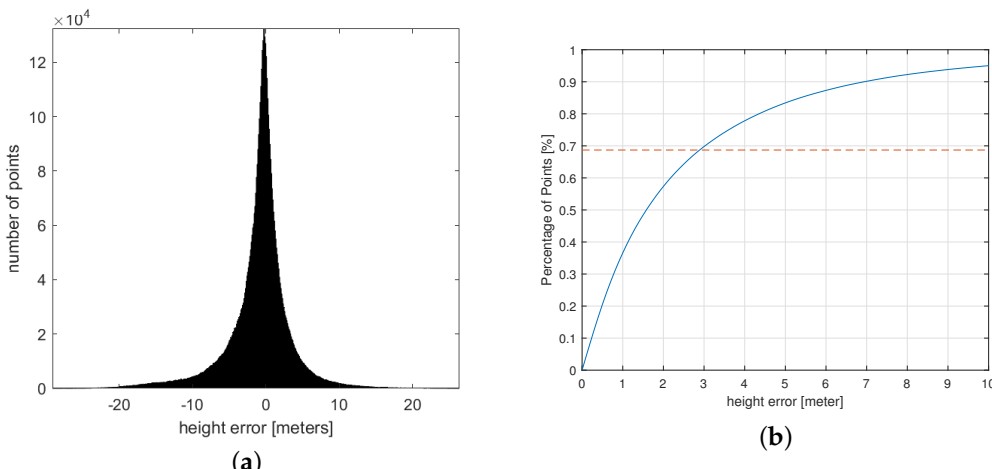

**Figure 11.** Evaluating height errors. (**a**) Histogram of height errors, (**b**) error-percentage curve.

At the end of this section, we add some additional comments to our accuracy evaluation methods and results. Firstly, the reconstructed 3D points from the stereo SAR method mostly correspond to pixels with strong gradients in the image domain, which generally have non-continuous height values in the image. Therefore, the evaluation results may have some bias for these points. Secondly, the ground truth data itself may also have errors, according to the accuracy of the adopted stereo SAR method. Additionally, the accuracy of 3D reconstruction on some continuous surfaces in the scene may not have been evaluated by this method. Therefore, the evaluation in this paper only provides a rough analysis of the 3D reconstruction accuracy of the proposed method.

The general conclusion in this section is that the proposed network can learn the 3D structural information of the scene with meter-level accuracy using CSAR amplitude images with a resolution of approximately 0.3 m.

## 5. Conclusions and Future Work

In this paper, we propose a new NeRF-like neural modeling framework for CSAR 3D inversion. Similar to NeRF, we use MLP as the solver instead of a convex optimization method. However, unlike NeRF, we propose a ray rendering strategy along the radar's elevation direction and choose not to explicitly model the transparency and visibility issues of the scene. We validate the 3D imaging performance of the proposed method with real-world data and demonstrate its potential applications and capabilities in generating new viewpoints and ortho-images.

As with NeRF, the final reconstructed scene corresponds to a 4D volume that is implicitly represented by the MLPs. The 3D reconstruction results can be projected onto any viewpoint, enabling us to generate new, uncollected observation data. This is an attempt to achieve CSAR 3D imaging using a neural network-based solver.

Currently, airborne SAR 3D imaging is generally achieved through TomoSAR methods, which require multiple-pass data collection or the equipment of a linear antenna array. In contrast, using a single-channel radar and circular trajectory for data collection is much more convenient. However, CSAR data are now more commonly only used as a video SAR, and there are few studies demonstrating the methods or results related to CSAR 3D image formation. One possible reason for this may be that the 3D coherent imaging pipeline for CSAR is too complex, requiring very precise motion compensation, sidelobe suppression methods, and dedicated optimization for anisotropic targets. In this paper, we propose a method for 3D imaging using neural networks and SAR amplitude images. The main significance of our approach is its simplicity in forming a 3D image. In addition to NeRF, there are many other differentiable rendering methods in the optical field whose rendering functions can be adapted with SAR imaging mechanisms, and introducing them into the radar field in the future may achieve better results than those from our method. We hope that our method can reignite scholars' enthusiasm for further studies of CSAR 3D imaging problems.

The scenario selected for our experiment was a school area, but SAR observation scenarios can be more intricate. For example, built-up areas with severe layover effects or targets like tanks with strong anisotropic scattering characteristics require further experimental verification to validate the effectiveness of this method. In addition, it is still an open question whether our algorithm can work effectively on lower-frequency bands such as the X-band. We encourage interested researchers to engage in relevant experimental studies in the future.

There are several points where the network in this paper can be improved. Firstly, our current model does not take into account the issue of occlusion and visibility caused by the shadowing effect. There are several emerging variants of NeRF that have addressed these issues, and these variants can be introduced in the future to improve the model in this paper. Secondly, the use of a small number of viewpoints to reconstruct a NeRF is also attractive, and it can be a future research topic for both airborne and space-borne SAR data. Thirdly, the network and the MLP solver used in our method may also be extend to other SAR 3D imaging problems (mainly the TomoSAR), which may improve the speed and quality of the 3D imaging or reduce the number of repeated flights required for tomographic 3D inversion.

**Author Contributions:** Methodology and writing, H.Z.; investigation, F.T., B.Y. and S.F.; supervision, W.H. and Y.L. All authors have read and agreed to the published version of the manuscript.

**Funding:** This research was supported by the National Natural Science Foundation of China under grant number 61860206013.

**Data Availability Statement:** Not applicable.

**Acknowledgments:** We sincerely thank the researchers from the Key Laboratory of IntelliSense Technology, Nanjing Research Institute of Electronics Technology for providing the necessary SAR data that supported the development of this algorithm. Their collaboration has been instrumental in achieving our research objectives, and we appreciate their dedication and expertise in advancing engineering research.

**Conflicts of Interest:** The authors declare no conflict of interest.

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
