# Peer review of "Circular SAR Incoherent 3D Imaging with a NeRF-Inspired Method"

_remotesensing, doi:10.3390/rs15133322_

Round 1

Reviewer 1 Report

Please see the above comments.

Reviewer 2 Report

This article introduces a novel NeRF-based method that leverages circular SAR image sequences to infer the 3D scattering power distribution of a scene. The technique can generate high-resolution 3D images using single-channel circular SAR data and create virtual SAR images while reconstructing the 3D shape of the imaged scene, as demonstrated in real-world experiments. I think the paper is valuable and worthy of publication in Remote Sensing after some minor revisions.

 1. line 142: ‘established’ → ‘establish’.

 2. line 155: Given that the authors used two coordinate systems in the article, I suggest that they replace 'g(x,y,z)' with 'g(x,y,z) (or g(u,v,w))' to reduce confusion for the reader in subsequent text.

 3. Figure 9. Ortho image of the scene : I suggest adding a colorbar on the right side of the image.

 4. Figure 11(a): The author should clarify the meaning of the y-axis or remove it.

 5. In the end of the experimental section, while the authors provide a detailed evaluation of the proposed method's accuracy, it would be helpful if they could also discuss the limitations of their approach. For example, what are the types of scenes or scenarios where the proposed network may not work well?

 Overall, I found the paper to be insightful and well-written with valuable contributions to the field. While there are a few minor revisions that need to be addressed, I believe that these can be easily implemented.

Reviewer 3 Report

Dear authors,

your work looks good, but I have three notes:

AD1: Please, describe how big were the errors of the measured positions by GNSS against the ideal circle trajectory. 

AD2: Please, insert also the statstics from the callibration measuremnts when you measured the position by the used GNSS when the receiver was on the static position.

AD3: Why you didn't realize more measurements for the validation? 

Best Regards and wishing good luck in your research.

Round 2

Reviewer 3 Report

All questions has been answered.

Wish you a good luck with your next research.

btw. In the future, when you will make answeres for more reviewers write them by the different colors (one color for every reviewer) and you can also copy the responds inside the response document.